# PLUG-AND-PLAY CONTROLLABLE GENERATION FOR DISCRETE MASKED MODELS

## ABSTRACT

This article makes discrete masked models for the generative modeling of discrete data controllable. The goal is to generate samples of a discrete random variable that adheres to a posterior distribution, satisfies specific constraints, or optimizes a reward function. This methodological development enables broad applications across downstream tasks such as class-specific image generation and protein design. Existing approaches for controllable generation of masked models typically rely on task-specific fine-tuning or additional modifications, which can be inefficient and resource-intensive. To overcome these limitations, we propose a novel plug-and-play framework based on importance sampling that bypasses the need for training a conditional score. Our framework is agnostic to the choice of control criteria, requires no gradient information, and is well-suited for tasks such as posterior sampling, Bayesian inverse problems, and constrained generation. We demonstrate the effectiveness of our approach through extensive experiments, showcasing its versatility across multiple domains, including protein design.

## 1 INTRODUCTION

Modeling complex discrete probability distributions in high-dimensional spaces is a crucial challenge across multiple domains in generative AI, including language, vision, audio, and biology. Among the various approaches, discrete masked models have emerged as powerful tools, offering robust solutions for generating and understanding discrete data. Notable examples include BERT for language modeling (Devlin et al., 2019), MaskGIT for image synthesis (Chang et al., 2022), DNABERT for DNA modeling (Ji et al., 2021; Zhou et al., 2023), the ESM series for protein generation (Rives et al., 2021; Lin et al., 2023; Hayes et al., 2024), and the more recent masked discrete diffusion models (see, e.g., Lou et al. (2024); Ou et al. (2024); Sahoo et al. (2024); Shi et al. (2024); Zheng et al. (2024)). These models typically learn the conditional distribution of each masked position given a partially masked data sequence, allowing for iterative decoding to generate a full sequence during inference.

In many practical applications of masked models, the objective extends beyond generating realistic samples from the data distribution to doing to in a *controlled* manner. This involves generating samples that align with specific constraints, conditions, or prompts, often by sampling from a conditional data distribution or maximizing a reward function (Zhang et al., 2023). Controlled generation is crucial in tasks such as (1) posterior sampling, where samples are drawn from a posterior distribution conditioned on observed data; (2) constrained generation, where the aim is to produce samples that meet predefined constraints. These applications highlight the growing need for generative models that not only capture the underlying data distribution, but also allow for flexible control over the generated outputs.

For controllable generation, it is desirable to establish a framework of *plug-and-play samplers* that allows for efficient sampling from the desired controlled distribution without requiring further training or fine-tuning of the underlying pre-trained model, given any suitable control criterion. This approach is computationally advantageous, as it avoids the costly and time-consuming process of retraining the model for each new task, making it a highly scalable and adaptable solution for real-world applications.

However, existing work on plug-and-play controllable generation primarily focused on the continuous domain, such as continuous diffusion models (Chung et al., 2023; Song et al., 2023; Huang et al.,

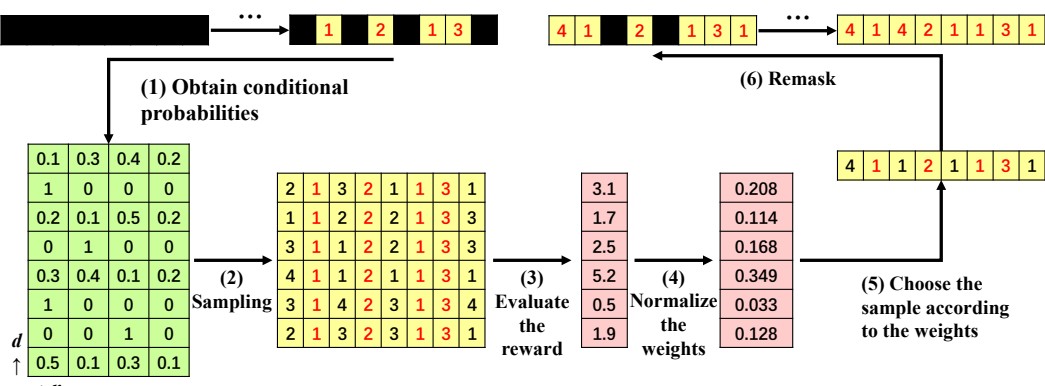

Figure 1: A demonstration of Algorithm 1 with vocabulary size $N = 4$, sequence length $D = 8$, and number of Monte Carlo estimate $K = 6$.

2024), and often requires the differentiability in the control criteria. In contrast, controllable generation for discrete generative models often relies on learning-based approaches (Dathathri et al., 2020; Nisonoff et al., 2024; Li et al., 2024), and to the best of our knowledge, there is no plug-and-play sampler in this domain.

In this paper, we address this gap by developing a plug-and-play framework for controllable generation using discrete masked models, eliminating the need for task-specific fine-tuning. Our algorithm operates through iterative unmasking and remasking. At each step, given the unmasked positions, we apply a mean-field approximation to estimate the conditional distribution of the masked positions, sample from it via Monte Carlo, and employ importance sampling to filter the most likely samples. We then remask a portion of the newly generated positions, and repeat this unmasking-remasking process for several times until all positions are unmasked. Since the complexity of querying the masked model to obtain conditional probabilities is typically much higher than querying the reward function in most of the real-world applications, the Monte Carlo estimation introduces minimal computational overhead, keeping the sampling and filtering process efficient. In our experiments, high-quality samples can be obtained with approximately 10 queries to the masked model and around 1000 Monte Carlo samples, demonstrating the effectiveness of our proposed algorithm.

In summary, our contributions are as follows:

- We tackle the problem of plug-and-play controllable generation for discrete masked models, introducing an efficient and economical paradigm for sampling from these models.

- We propose a novel framework based on mean-field approximation of multi-dimensional discrete distributions and iterative masking-unmasking. This fine-tuning-free approach enables sample generation that satisfies control criteria for any (potentially non-differentiable) reward functions. To the best of our knowledge, this is the first plug-and-play controllable sampler for discrete masked models.

- We demonstrate the versatility of our method through multiple experiments, including sampling sequence of integers with equality constraint and designing protein sequences, highlighting its adaptability and effectiveness across diverse tasks.

**Notations.** The indicator function $1_A$ of a statement $A$ is 1 if $A$ is true and 0 if otherwise. We will use superscripts for indexing a vector, e.g., $x = (x^1, \ldots, x^D)$. Given $x \in \mathbb{R}^D$ and a set $\Omega \subset \{1, \ldots, D\}$, the slicing $x^\Omega$ is defined as $(x^d : d \in \Omega)$. Given a partition $(\Omega_1, \Omega_2)$ of $\{1, \ldots, D\}$ (i.e., $\Omega_1, \Omega_2$ are disjoint and their union is $\{1, \ldots, D\}$), and two vectors $x = (x^d : d \in \Omega_1)$ and $y = (y^d : d \in \Omega_2)$, their concatenation $x \oplus y$ is a $D$-dimensional vector whose $d$-th entry is $x^d 1_{d \in \Omega_1} + y^d 1_{d \in \Omega_2}$. Finally, $f(x) \propto_x g(x)$ means that $f(x) = c \cdot g(x)$ for some constant $c > 0$ that does not depend on $x$.

## 2 PRELIMINARIES AND PROBLEM FORMULATION

### 2.1 DISCRETE MASKED MODELS

**Discrete masked models** (or simply **masked models**) are designed to learn the distribution of sequential discrete data. Let $\{1, \ldots, N\}$ represent a vocabulary of $N$ tokens, and consider a random sequence $X = (X^1, \ldots, X^D) \in \{1, \ldots, N\}^D$ of length $D$. For example, $X$ can be a sentence with $D$ words, a protein composed of $D$ amino acids, or a discrete latent representation of images tokenized by a vector quantized variational autoencoder (VQ-VAE) (van den Oord et al., 2017). To learn the probability distribution $p$ of $X$ given i.i.d. samples of $X \sim p$, masked models are trained by randomly replacing certain positions in the sequence with a masked token $\mathbf{M}$, and learning the probability of the masked positions conditional on the unmasked portion of the sequence.

Throughout this paper, for a partially observed sequence $x \in \{1, \ldots, N, \mathbf{M}\}^D$, we use $\Omega := \{d : x^d \neq \mathbf{M}\}$ and $\mathcal{M} := \{d : x^d = \mathbf{M}\}$ to denote the unmasked and masked positions, respectively. Formally, a masked model $\widehat{p}$ takes such a sequence as input and output a matrix $\widehat{p}(x) \in \mathbb{R}_{\geq 0}^{D \times N}$ that approximates the following probability distribution:

$$\widehat{p}(x)_{d,n} \approx p(X^d = n | X^\Omega = x^\Omega).$$

The rows in $\widehat{p}$ corresponding to the unmasked positions in $\Omega$ are trivial, as the probabilities are either $0$ or $1$. To learn the remaining entries, masked models are typically trained by minimizing the cross-entropy loss:

$$\min_{\widehat{p}} \mathbb{E}_{X \sim p} \mathbb{E}_{\text{random subset } \Omega \subset \{1, \ldots, D\}} \left( -\log \sum_{d \notin \Omega} \widehat{p} \left( X^\Omega \oplus \mathbf{M} \right)_{d, X^d} \right),$$

where $X^\Omega \oplus \mathbf{M}$ represents the sequence obtained by replacing all entries not in $\Omega$ with $\mathbf{M}$.

During inference, a direct approach is to initialize with the fully-masked sequence, select an order $\sigma$ (a permutation of $\{1, \ldots, D\}$), and autoregressively sample $X^{\sigma(t)}$ based on $X^{\sigma(<t)}$ for $t = 1, 2, \ldots, D$. However, this scheme requires $D$ queries to the masked model, which is inefficient for long sequences. To balance accuracy and efficiency, instead of unmasking one position at a time, one may introduce a decreasing function $\gamma : [0, 1] \to [0, 1]$ known as an **unmasking schedule**, which determines the number of remaining masked tokens at each step (Chang et al., 2022). At the $t$-th step ($t = 1, 2, \ldots, T$) out of $T$ total steps, given $x$ with observed positions $\Omega$, the following steps are implemented:

1. Predict the probability of the masked positions $p(X^d = \cdot | X^\Omega = x^\Omega) \approx \widehat{p}(x)_{d,\cdot}$, $d \notin \Omega$ using the masked model.

2. Independently sample the masked values $x^d \sim \widehat{p}(x)_{d,\cdot}$, $d \notin \Omega$.

3. Remask $\lfloor \gamma(t/T)D \rfloor$ newly-generated positions with the lowest predicted probability $\widehat{p}(x)_{d,x^d}$.

Such steps will also be utilized in the design of our algorithm.

There are several equivalent formulation of masked models. First, as demonstrated in the sampling procedure outlined above, they can be interpreted as any-order autoregressive models (Hoogeboom et al., 2022; Shih et al., 2022), where the joint distribution is factorized using any arbitrary order $\sigma$, and the model learns the conditional distributions $p(X^{\sigma(d)} | X^{\sigma(<d)})$, $d \in \{1, \ldots, D\}$. Second, recent research (Ou et al., 2024; Sahoo et al., 2024; Shi et al., 2024; Zheng et al., 2024) has shown that masked models are equivalent to the masked discrete diffusion model, which involves reversing a continuous-time Markov chain that transform any data distribution into the distribution concentrated on the sequence with all masked states.

### 2.2 CONTROLLABLE GENERATION

This paper focuses on the following task of **controllable generation** for masked models: given a pretrained masked model $\widehat{p}(\cdot)$ that generates from a data distribution $p$, sample from a modified

distribution

$$q(x) = \frac{1}{Z} r(x) p(x), \; x \in \{1, \dots, N\}^D,$$

where $r(x)$ is a non-negative **reward function**, and the normalizing constant $Z = \sum_x r(x) p(x)$ is unknown. This formulation encompasses various applications, including:

- **Posterior sampling for Bayesian inference**: Suppose $X \sim p(x)$ and we have a conditional distribution $p(y|x)$ for a related random variable $Y$, where $Y|X = x \sim p(y|x)$ serves as the reward function. Given an observation $Y = y$, the posterior distribution of $X$ is $p(x|y) \propto_x p(y|x)p(x)$. For instance, if $X$ is a tokenized image and $Y$ is its corresponding class label given by a classifier, then sampling from $X|Y = y$ would generate an image belonging to a specific class $y$.
- **Constrained generation**: Given a specific subset $S \subset \{1, \dots, N\}^D$ of interest, we define the reward function as the indicator function $r(x) = 1_{x \in S}$. In this case, $q$ represents the distribution $p$ truncated to the set $S$. For example, $X$ represents DNA sequences, and $S$ is the set of DNA sequences whose percentage of A is less than $30\%$.

We also assume that evaluating the reward function $r(\cdot)$ is significantly less computationally expensive than querying the mask model $\widehat{p}(\cdot)$, which is a common scenario in most of the real-world applications, and serves as an important starting point for our algorithm's design.

## 3 CONTROLLABLE GENERATION FOR MASKED MODELS

In this section, we present our framework for plug-and-play controllable generation of masked models. We begin by drawing insights from the plug-and-play conditional generation for continuous diffusion models in Section 3.1, and then propose our novel algorithms tailored specifically for discrete masked models in Section 3.2.

### 3.1 EXISTING PLUG-AND-PLAY SAMPLERS FOR CONTINUOUS DIFFUSION MODELS

In continuous diffusion model, the data distribution is $X_0 \sim p(x_0)$, and a diffusion process $(X_t \sim p_t)_{t \in [0,T]}$ transforms data to noise following $X_t|X_0 = x_0 \sim \mathcal{N}(x_0, \sigma_t^2 I)$. Leveraging the data samples and the diffusion process, one can learn the score function $\nabla_{x_t} \log p_t(x_t)$ for all $t > 0$. Given a condition variable $Y$ with distribution $p(y|x_0)$ conditional on $X_0 = x_0$, the task of sampling from the posterior distribution $p(x_0|y)$ boils down to estimating $\nabla_{x_t} \log p_t(y|x_t)$ for all $t > 0$, where $p_t(y|x_t)$ is the predicted distribution of $Y$ given *noisy* sample $X_t = x_t$.

To estimate this quantity in a training-free way, Chung et al. (2023) observed that $p_t(y|x_t) = \mathbb{E}_{p(x_0|x_t)} p(y|x_0)$, and proposed to approximate the unknown distribution $p(x_0|x_t)$ by the point mass at $\widehat{x}_0(x_t) := \mathbb{E}(X_0|X_t = x_t) = x_t + \sigma_t^2 \nabla_{x_t} \log p_t(x_t)$, resulting in $p_t(y|x_t) \approx p(y|\widehat{x}_0(x_t))$. Song et al. (2023) later proposed a Gaussian approximation centered at $\widehat{x}_0(x_t)$ with a suitable variance, estimating the expectation by the empirical mean over i.i.d. Gaussian samples. Finally, one step of backward propagation yields the gradient of $\log p(x_0|x_t)$ with respect to $x_t$. Throughout this process, only one query to the score model $\nabla_{x_t} \log p_t(x_t)$ is required, while the conditional probability $p(y|x_0)$ is generally easy to obtain, minimizing computational overhead.

To sum up, the key ingredient of this approach is the approximation of the distribution $p(x_0|x_t)$, which involves predicting the clean data $X_0$ given a noisy sample $X_t = x_t$.

### 3.2 CONTROLLABLE GENERATION FOR MASKED MODELS

We first continue the exploration of conditional generation, and then extend our methodology to general controllable generation problems.

In the discrete case, unlike the Gaussian noise $X_t|X_0 = x_0 \sim \mathcal{N}(x_0, \sigma_t^2 I)$, we now have a masking process $X_t|X_0 = x_0$ obtained by independently masking each position with a probability that depends on $t$. As the required entity is not the score function here, we cannot directly apply the continuous diffusion model's strategy. Moreover, the discrete state space lacks a Gaussian distribution,

and the posterior mean $\mathbb{E}(X_0|X_t = x_t)$ is meaningless. However, a simple yet effective alternative exists: a mean-field approximation.

Our goal is to sample from $p(x_0|x_t, y)$. By conditional independence of $X_t$ and $Y$ given $X_0$,

$$p(x_0|x_t, y) \propto_{x_0} p(x_0, x_t, y) = p(x_0)p(x_t|x_0)p(y|x_0) = p(x_0|x_t)p(x_t)p(y|x_0)$$
$$\implies p(x_0|x_t, y) \propto_{x_0} \mathbb{E}_{p(x_0|x_t)} p(y|x_0).$$

As $X_t$ is by independently masking each position in $X_0$, we only need to predict the conditional probability of the masked positions in $x_t$ given the observed ones. As the masked model only predicts one-dimensional probability distributions, a straightforward approach is to iteratively un-masking one position at a time, requiring $|\mathcal{M}|$ queries to the masked model.

To avoid such computational overhead, we employ a **mean-field approximation**, i.e., assume that conditional on the observed part $x^\Omega$ of a sequence $x$, each remaining dimension in $\mathcal{M}$ is independent. Formally,

$$p(X^{\mathcal{M}} = u|X^\Omega = x^\Omega) \approx \prod_{d \in \mathcal{M}} p(X^d = u^d|X^\Omega = x^\Omega),$$

which only requires one query to the masked model. While mean-field approximation introduces some error, it is effective for approximating multidimensional distributions when dependencies be-tween different positions are relatively weak. Similar ideas have been incorporated to the sampling algorithm of MaskGIT (Chang et al., 2022) and the backward sampling of Continuous-Time Markov Chain (Lou et al., 2024; Ou et al., 2024; Zheng et al., 2024).

Building upon our insights into conditional generation, we are ready to present a solution to the general problem of controlled generation. Based on the problem settings in Section 2.2, we now illustrate our method for sampling from $q$ in a plug-and-play manner.

Suppose we have a partially observed sequence $x$ during the generation process, with observed and masked positions $\Omega$ and $\mathcal{M}$. We can calculate the conditional distribution of masked positions given the observed ones under $q$ as follows: for all $u \in \{1, \dots, N\}^{|\mathcal{M}|}$,

$$\begin{aligned}
q(X^{\mathcal{M}} = u|X^\Omega = x^\Omega) &\propto_u q(X^{\mathcal{M}} = u, X^\Omega = x^\Omega) \\
&\propto_u r(x^\Omega \oplus u)p(X^{\mathcal{M}} = u, X^\Omega = x^\Omega) \\
&\propto_u r(x^\Omega \oplus u)p(X^{\mathcal{M}} = u|X^\Omega = x^\Omega) \\
&\approx r(x^\Omega \oplus u) \prod_{d \in \mathcal{M}} p(X^d = u^d|X^\Omega = x^\Omega),
\end{aligned} \tag{1}$$

where the last line is obtained by mean-field approximation. As the normalizing constant of this distribution is unknown, one way to sample a $u$ from this distribution is by leveraging importance sampling. Let us first recall the following lemma.

**Lemma 1** (Importance Sampling). *Suppose two probability masses or densities $p, q$ are related through $q(x) = \frac{1}{Z}r(x)p(x)$. Then, with $x_1, \dots, x_K$ i.i.d. samples from $p$, one can approximate $q$ with the following weighted empirical distribution:*

$$q(x) \approx \widetilde{q}(x) := \sum_{k=1}^{K} q_k \delta_{x_k}(x), \; where \; q_k = \frac{r(x_k)}{\sum_{j=1}^{K} r(x_j)}.$$

*Proof.* Since $p(x) \approx \widetilde{p}(x) := \frac{1}{K} \sum_{k=1}^{K} \delta_{x_k}(x)$, we have the following approximation of $q$:

$$q(x) = \frac{r(x)p(x)}{\mathbb{E}_p r} \approx \frac{r(x)\widetilde{p}(x)}{\mathbb{E}_{\widetilde{p}} r} = \frac{r(x) \sum_{k=1}^{K} \delta_{x_k}(x)}{\sum_{k=1}^{K} r(x_k)} = \widetilde{q}(x).$$

$\square$

Thanks to Lemma 1, we can approximately sample from the distribution in Equation (1) by indepen-dently sampling each dimension $d \in \mathcal{M}$ conditional on the observed sequence $x^\Omega$ and then obtain their corresponding weights, using which we can sample a $u$.

---

**Algorithm 1:** Plug-and-play controllable sampler of discrete masked models

---

**Input:** Vocabulary size $N$, length of sequence $D$, reward function $r(\cdot)$, masked model $\widehat{p}(\cdot)$, number of unmasking steps $T$, unmasking schedule $\gamma : [0,1] \to [0,1]$, number of Monte Carlo samples $K$.

**Output:** An approximate sample $x$ from the distribution $q(x) \propto r(x)p(x)$.

1 Initialize $x = (\mathbf{M}, \ldots, \mathbf{M})$, the mask positions $\mathcal{M} = \{1, \ldots, D\}$, and the observed positions $\Omega = \varnothing$;

2 **for** $t = 1$ **to** $T$ **do**

3      Compute the probabilities $\{\widehat{p}(x)_{d,n} \approx p(X^d = n | X^\Omega = x^\Omega) : d \in \mathcal{M}, \ n \in \{1, \ldots, N\}\}$ from the masked model;

4      **for** $d \in \mathcal{M}, \ k \in \{1, \ldots, K\}$ *(in parallel)* **do**

5          Independently sample $u^d_{(k)} \sim \widehat{p}(x)_{d,\cdot}$;

6      **end**

7      **for** $k \in \{1, \ldots, K\}$ *(in parallel)* **do**

8          Assign the sample $u_{(k)} = (u^d_{(k)} : d \in \mathcal{M})$ with a weight $w_{(k)} = r(x^\Omega \oplus u_{(k)})$;

9      **end**

10      **for** $k \in \{1, \ldots, K\}$ *(in parallel)* **do**

11          Normalize the weights $w_{(k)}$ by $w_{(k)} \leftarrow \frac{w_{(k)}}{\sum_{j=1}^{K} w_{(j)}}$;

12      **end**

13      Obtain one sample of $x^\mathcal{M}$ via selecting $u_{(k)}$ with probability $w_{(k)}$;

14      Remask $\lfloor \gamma(t/T)D \rfloor$ positions in $x^\mathcal{M}$ according to a defined rule (e.g., uniform remasking: sample a subset $\mathcal{M}_0$ of $\mathcal{M}$ with $\lfloor \gamma(t/T)D \rfloor$ elements uniformly at random, and remask the positions $x^d, \ d \in \mathcal{M}_0$);

15      Update the masked positions $\mathcal{M} \leftarrow \{d : x^d = \mathbf{M}\}$ and the observed positions $\Omega \leftarrow \mathcal{M}^\complement$;

16 **end**

17 **return** $x$.

---

Although this process generates a full sequence $x$, its alignment to the target distribution $q$ may be sub-optimal due to the lost of dependencies in mean-field approximation, especially when $|\mathcal{M}|$ is large (i.e., many positions are unmasked in a row). To address this, we further introduce a re-masking step as discussed in Section 2.1: remask some of positions in $\mathcal{M}$ according to a remasking schedule $\gamma$ (e.g., by sampling a random subset of $\mathcal{M}$), so that in the $t$-th step among the $T$ total steps ($t = 1, \ldots, T$), after remasking, the remaining number of masked tokens is $\lfloor \gamma(t/T)D \rfloor$. In our experiments, we found that this uniform remasking strategy performs well. However, exploring more advanced remasking strategies is a promising area for future research.

We summarize the entire procedure in Algorithm 1. Notably, our algorithm can be easily extended for the inpainting task, i.e., given a subset $\overline{\Omega}$ of $\{1, \ldots, D\}$ that has known values $x^{\overline{\Omega}}$, we aim to sample the remaining positions according to $q(X^{\overline{\Omega}^\complement} = \cdot | X^{\overline{\Omega}} = x^{\overline{\Omega}})$. This modified version of the algorithm is provided in Algorithm 2.

## 4 RELATED WORK

**Conditional generation based on guidance.** As an important task in controllable generation, conditional generation aims to generate a random variable $X \sim p(x)$ (e.g., image) given another random variable $Y$ known as the condition (e.g., the text description of an image). A popular approach is guidance: for example, in continuous diffusion model, the classifier guidance (Dhariwal & Nichol, 2021) learns a classifier $p(y|x)$ that predicts the condition $Y$ given a *noisy* sample of $X$ and leverages its information to generate $X$ conditional on $Y = y$. Classifier-free guidance (Ho & Salimans, 2022) trains a score model that approximates both the conditional and unconditional score functions, using a combination of them for conditional generation. These approaches can be extended to the discrete diffusion model (see Nisonoff et al. (2024)).

**Controllable generation for discrete generative models.** The study of controllable generation is an emerging area in language modeling (see, e.g., Zhang et al. (2023) for a review). A notable work, Dathathri et al. (2020), proposed applying gradient updates to the key-value cache in transformers, a task-agnostic approach but requiring fine-tuning during inference. For diffusion models, a recent work Li et al. (2024) introduced soft value-based decoding, a derivative-free algorithm that requires pre-sampled trajectories $x_0, x_1, \ldots, x_T$ of discrete diffusion model to estimate a conditional expectation. This method does not exploit the special properties of the masking process. To the best of our knowledge, there are no fine-tuning-free samplers for controllable generation in discrete masked models.

## 5 EXPERIMENTAL RESULTS

### 5.1 TOY EXPERIMENT ON SAMPLING EQUALITY-CONSTRAINED SEQUENCES

We first apply our controllable generation framework to constrained sampling, demonstrating its outstanding efficiency in challenging tasks where the constraint set is extremely sparse.

Consider the state space $\{1, \ldots, N\}^D$ comprising sequences of integers $x = (x^1, \ldots, x^D)$, where we fix the sequence length $D$ to 10. Let the unconditional distribution be the uniform distribution. In this case, the masked model has a closed form, requiring no training:

$$\widehat{p}(x)_{d,n} = p(X^d = n | X^\Omega = x^\Omega) = \begin{cases} 1/N, & d \in \mathcal{M} \\ \delta_{x^d, n}, & d \in \Omega \end{cases}.$$

We study sampling from the uniform distribution restricted to the following set:

$$S_N = \left\{ x \in \{1, \ldots, N\}^D : \phi(x) := x^1 - x^2 x^3 - x^4 + x^5 x^6 x^7 + x^8 + x^9 - x^{10} = 0 \right\}.$$

The ratio between the cardinalities of $S_N$ and the entire state space $\{1, \ldots, N\}^D$ is extremely low: approximately $1.07\%$ when $N = 10$, $0.20\%$ when $N = 20$, and $0.07\%$ when $N = 30$.

We use Algorithm 1 to sample from the uniform distribution constrained on $S_N$, and present the results in Figure 2. We observe that the number of Monte Carlo samples, $K$, significantly impacts the quality of generated samples, while the number of steps of unmasking $T$ has a less pronounced effect, likely due to the short sequence length. Further experimental details are provided in Appendix B.1.

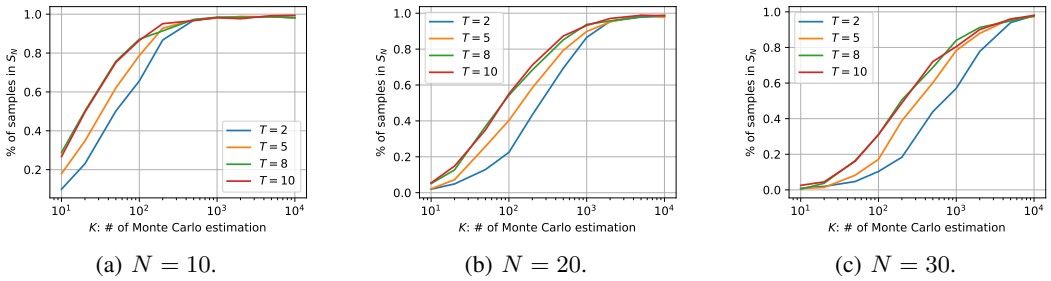

(a) $N = 10$.      (b) $N = 20$.      (c) $N = 30$.

Figure 2: Result for sampling equality-constrained sequences.

### 5.2 CONTROLLABLE PROTEIN GENERATION

We employ ESM3 (Hayes et al., 2024) as the underlying protein generation model, which is a pioneering generative model for protein, capable of reasoning simultaneously over multiple modalities including sequence, structure, and function, achieving state-of-the-art performance in multiple protein generation tasks. In our experiments, we focus on generation in the *sequence* domain, and fix the length of sequence to 50. The vocabulary considered is the set of 20 standard amino acids[1]. We will consider three tasks of controlled generation: generating hydrophilic and hydrophobic proteins,

---

[1]They are: A, C, D, E, F, G, H, I, K, L, M, N, P, Q, R, S, T, V, W, Y.

sampling proteins with a propensity for alpha-helices, and protein inpainting for higher percentage of alpha-helices. In all tasks, we impose a constraint based on the instability index (Guruprasad et al., 1990), a positive number that estimates the stability of a protein sequence. A protein with value between 0 and 40 is predicted as stable. To evaluate all the required metrics of protein sequences, we use the Biopython package (Cock et al., 2009), a widely used tool in computational molecular biology.

**Design of reward function.** To begin with, we first introduce a flexible way of designing the reward function $r(\cdot)$ for a given task. Suppose we have $M$ metrics $m_i : \{1, \ldots, N\}^D \to \mathbb{R}, i = 1, 2, \ldots, M$ that we are interested in, and our constraint is expressed as

$$\bigcap_{i=1}^{M} \{x : m_i(x) \in A_i\},$$

where $A_i \subset \mathbb{R}$ is an interval. We propose the following reward function:

$$r(x) = \exp\left(-\sum_{i=1}^{M} w_i \operatorname{dist}(m_i(x), A_i)^{\alpha_i}\right),$$

where $\operatorname{dist}(a, A)$ is the distance from $a \in \mathbb{R}$ to the interval $A \subset \mathbb{R}$, $w_i > 0$ is the weight of the $i$-th constraint ($m_i(x) \in A_i$), and $\alpha_i > 0$ determines the shape of penalty (e.g., linear or quadratic).

**Sampling hydrophilic/hydrophobic proteins.** Solubility is one of the key traits of protein and it is a joint consequence of amino acid sequence composition as well as 3D structure. Protein solubility is often quantified through the hydropathy index, with high hydropathy values indicating water-repelling (hydrophobic) and low hydropathy value indicating water-attracting (hydrophilic). Designing hydrophilic or hydropathic proteins have numerous important applications, yet the tasks also pose unique challenges (Qing et al., 2022). Low hydropathy value is often desirable for therapeutic purposes as it's central to the protein expression and purification process. Hydrophilicity is also critical for a longer circulation time in the human bloodstream, potentially indicating a better therapeutic efficacy (Garidel, 2013). Proteins with high hydropathy value are often designed for transmembrane proteins, important examples of which include cell surface receptors. Transmembrane proteins are often targets of drugs, especially receptors like G-protein-coupled receptors. Designing these proteins can help create more effective disease treatments by improving the capability to modulate cellular signals (Yang et al., 2021).

In the following, we consider the challenging task of generating hydrophilic and hydrophobic protein sequences. We set the length of protein sequence to 50, and use GRAVY (Grand Average of Hydropathy) value (Kyte & Doolittle, 1982) to quantify the hydropathy level of the generated sequence, which is defined as the average hydropathy value of a peptide or protein. Negative GRAVY value indicates hydrophilic, while positive one means hydrophobic.

We compare the GRAVY values of the samples generated by ESM3 model in both uncontrolled and controlled way in Table 1. For controlled generation, we fix the hyperparameters $w_2$ and $A_2$ for promoting low instability index, and experimented with multiple choices of $w_1$ and $A_1$ (see further details in Appendix B). The table presents the best generation result. Using our proposed controllable generation method, we successfully sample protein sequences with the desired values of GRAVY while maintaining protein stablility. We also observe a significant reduction in standard deviation among the controlled generated samples, highlighting the reliability of our method.

Table 1: Controlled generation for high and low GRAVY values. The metrics are presented in the form of mean $\pm$ std.

| Task | $m_1 = $ GRAVY | $m_2 = $ instability | GRAVY | Instability $\downarrow$ |
|---|---|---|---|---|
| Uncontrolled | / | / | $-0.207 \pm 0.552$ | $41.467 \pm 17.906$ |
| Controlled, high GRAVY | $w_1 = 30, \alpha_1 = 1,$ $A_1 = [1, \infty)$ | $w_2 = 5, \alpha_2 = 2,$ $A_2 = [0, 40]$ | $1.209 \pm 0.223$ | $25.176 \pm 10.094$ |
| Controlled, low GRAVY | $w_1 = 35, \alpha_1 = 1,$ $A_1 = (-\infty, -1]$ | $w_2 = 5, \alpha_2 = 2,$ $A_2 = [0, 40]$ | $-1.261 \pm 0.260$ | $31.569 \pm 8.204$ |

**Sampling alpha-helix-rich protein.** Proteins fold naturally into unique three-dimensional structures based on their amino acid sequence composition. This spatial conformation further determines

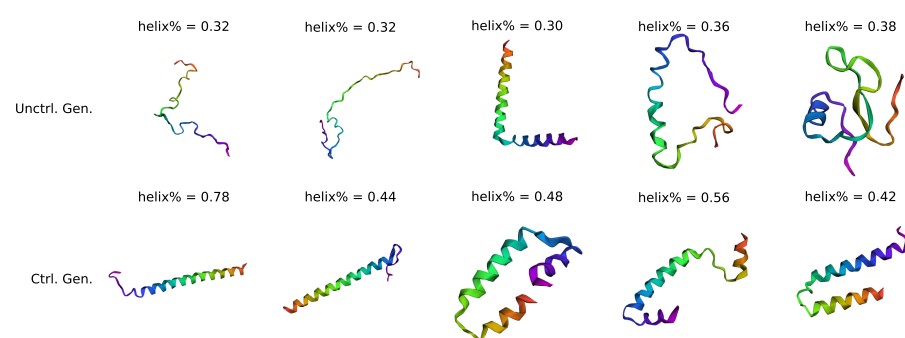

Figure 3: Structure of protein sequences predicted by ESM3. The upper row are the uncontrolled generated sequences, and the lower row are the controlled generated sequences. The sequences are randomly chosen.

its molecular and cellular function. Among the possible spatial structures, alpha helix is one of the most common secondary structures in protein, where the amino acids in the polypeptide chain form into a coil (helix) through hydrogen bonding. Particularly, designing proteins rich in alpha helices is of special interest for engineering certain protein functions (Kortemme, 2024), such as binding and self-assembly (Sakuma et al., 2024). These functions are central to applications such as therapeutic protein discovery (Walsh & Jefferis, 2006) and alpha-helical nanofiber design (Zhang, 2003).

Motivated by the versatile purpose of protein with rich alpha-helix structures, in the following, we focus on generating protein sequences with secondary structure being all or predominant alpha-helices. We compare the metrics of the generated proteins by our proposed method with those generated by the unconditional model in Table 2. Our controlled generation method successfully produces sequences with a high predicted percentage of alpha-helices (helix%). To verify the structural accuracy of the generated sequences, we use the folding algorithm provided by ESM3 to predict their 3D structures given the sequences. The result is displayed in Figure 3, where we randomly sample five generated sequences from both the uncontrolled and controlled generated sets and predict their structure. The controlled generated samples exhibit a higher frequency of alpha-helices in the predicted structure, confirming the effectiveness of our algorithm. For further details on the figures, please refer to Appendix B.3.

We also investigate the influence of the hyperparameters $w_i$ and $A_i$ in the design of reward function $r(\cdot)$. Our empirical findings suggest an optimal choice for these parameters, providing valuable insights into designing reward function for general controlled generation problems. Additional details are presented in Appendix B.2.

Table 2: Controlled generation for high alpha-helix percentage. The metrics are presented in the form of mean $\pm$ std.

| Task | $m_1 = $ helix% | $m_2 = $ instability | Helix% $\uparrow$ | Instability $\downarrow$ |
|---|---|---|---|---|
| Uncontrolled | / | / | $0.315 \pm 0.072$ | $41.467 \pm 17.906$ |
| Controlled | $w_1 = 50, \alpha_1 = 1,$ $A_1 = [0.8, \infty)$ | $w_2 = 5, \alpha_2 = 2,$ $A_2 = [0, 40]$ | $0.600 \pm 0.116$ | $27.387 \pm 12.025$ |

**Protein inpainting.** We consider the following inpainting problem: using a protein chain with high beta-sheet percentage as a prompt to generate the remaining positions in a controlled manner to maximize the percentage of alpha-helices. In particular, the prompt is chosen as a 35-amino-acid slice from the SM-related protein of P. Abyssi (PDB ID: 1H64), chain A, which consists of 71 amino acids and has a secondary structure composed almost entirely of beta-sheets. The cyan motif in the subfigure wrapped in red box in Figure 4 highlights the prompt. We use Algorithm 2 to generate alpha-helix-rich proteins based on this prompt, and display the predicted 3D structure

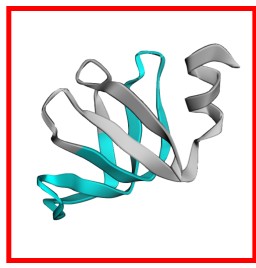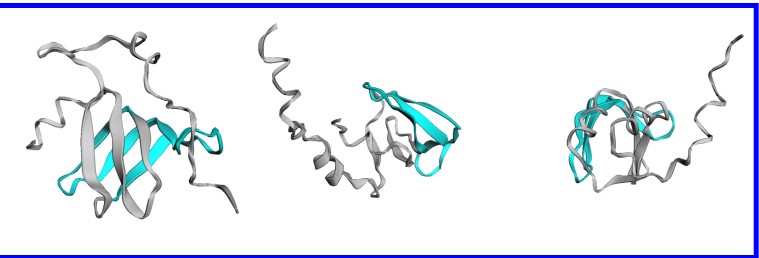

Figure 4: Visualization of the generated sequence from protein inpainting.

of three randomly generated sequences in the blue box in Figure 4, demonstrating the presence of alpha-helices in the generated portions. Further experimental details can be found in Appendix B.4.

## 6 CONCLUSION AND FUTURE WORK

In this paper, we study the task of controllable generation for discrete masked models and introduce an efficient paradigm for plug-and-play sampling. Our approach is based on mean-field approximation and iterative masking and remasking, demonstrating promising potential for real-world applications. Future research directions include exploring alternative remasking strategies beyond uniform remasking, rigorously analyzing the error bounds of our method, and extending its application to other domains such as image, molecule, and DNA.

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

# A    ALGORITHM FOR INPAINTING

See Algorithm 2. Its difference with Algorithm 1 is marked in blue.

---

**Algorithm 2:** Plug-and-play controllable sampler of discrete masked models, the inpainting version.

---

**Input:** Vocabulary size $N$, length of sequence $D$, reward function $r(\cdot)$, masked model $\widehat{p}(\cdot)$, number of unmasking steps $T$, unmasking schedule $\gamma : [0, 1] \rightarrow [0, 1]$, number of Monte Carlo samples $K$, inpainted indices $\overline{\Omega} \subset \{1, \ldots, D\}$, inpainted values $x^{\overline{\Omega}} \in \{1, \ldots, N\}^{|\overline{\Omega}|}$.

**Output:** An approximate sample $x$ from the distribution $q(x) \propto r(x)p(x)$.

1 Initialize $x = x^{\overline{\Omega}} \oplus \mathbf{M}$, the mask positions $\mathcal{M} = \overline{\Omega}^{\complement}$, and the observed positions $\Omega = \overline{\Omega}$;

2 **for** $t = 1$ **to** $T$ **do**

3     Compute the probabilities $\{\widehat{p}(x)_{d,n} \approx p(X^d = n | X^\Omega = x^\Omega) : d \in \mathcal{M},\ n \in \{1, \ldots, N\}\}$ from the masked model;

4     **for** $d \in \mathcal{M},\ k \in \{1, \ldots, K\}$ *(in parallel)* **do**

5         Independently sample $u^d_{(k)} \sim \widehat{p}(x)_{d,\cdot}$;

6     **end**

7     **for** $k \in \{1, \ldots, K\}$ *(in parallel)* **do**

8         Assign the sample $u_{(k)} = (u^d_{(k)} : d \in \mathcal{M})$ with a weight $w_{(k)} = r(x^\Omega \oplus u_{(k)})$;

9     **end**

10     **for** $k \in \{1, \ldots, K\}$ *(in parallel)* **do**

11         Normalize the weights $w_{(k)}$ by $w_{(k)} \leftarrow \frac{w_{(k)}}{\sum_{j=1}^K w_{(j)}}$;

12     **end**

13     Obtain one sample of $x^{\mathcal{M}}$ via selecting $u_{(k)}$ with probability $w_{(k)}$;

14     Remask $\lfloor \gamma(t/T)(N - |\overline{\Omega}|) \rfloor$ positions in $x^{\mathcal{M}}$ according to a defined rule (e.g., uniform remasking: sample a subset $\mathcal{M}_0$ of $\mathcal{M}$ with $\lfloor \gamma(t/T)(N - |\overline{\Omega}|) \rfloor$ elements uniformly at random, and remask the positions $x^d,\ d \in \mathcal{M}_0$);

15     Update the masked positions $\mathcal{M} \leftarrow \{d : x^d = \mathbf{M}\}$ and the observed positions $\Omega \leftarrow \mathcal{M}^{\complement}$;

16 **end**

17 **return** $x$.

---

# B    SUPPLEMENTARY EXPERIMENTAL RESULTS

## B.1    IMPLEMENTATION DETAILS OF THE TOY EXAMPLE

We first demonstrate how we choose the reward function $r(\cdot)$. Recall that the equality constraint is $\phi(x) = 0$. Thus, we propose to use the reward function $r(x) = \exp(-w \max(|\phi(x)|, m))$, where the weight $w > 0$ and the truncation threshold $m > 0$. We choose $w = 5$ and $m = 10$ throughout all of our experiments.

We fix the remasking schedule $\gamma(t) = \cos\left(\frac{\pi}{2} r\right)$ as suggested by Chang et al. (2022).

The values of $K$ experimented in Figure 2 are $10, 20, 50, 100, 200, 500, 1000, 2000, 5000, 10000$, while the values of $T$ experimented are $2, 5, 8, 10$.

## B.2    TUNING THE HYPERPARAMETERS FOR CONTROLLED PROTEIN GENERATION

In this section, we study how the hyperparameters $w_i$ and $A_i$ in the reward function influences the quality of controllable generation. In particular, we focus on the task of sampling alpha-helix rich protein, fix the hyperparameters for $m_2 =$ instability, and vary the choice of $w_1, A_1$ while fixing $\alpha_1 = 1$ throughtout the experiments. We experiment $w_1 \in \{10, 15, \ldots, 45, 50\}$ and $A_1 = [a_1, \infty)$ for $a_1 \in \{0.5, 0.6, \ldots, 1.3, 1.4\}$, and for each pair of $(w_1, A_1)$, generate 16 protein sequences and

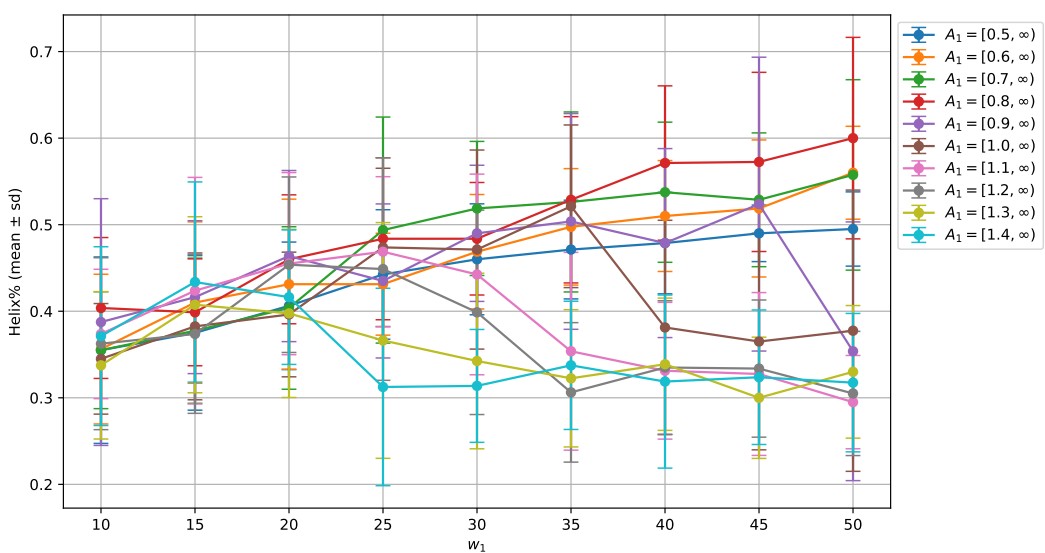

Figure 5: Influence of $w_1$ and $A_1$ on helix% in sampling alpha-helix rich proteins.

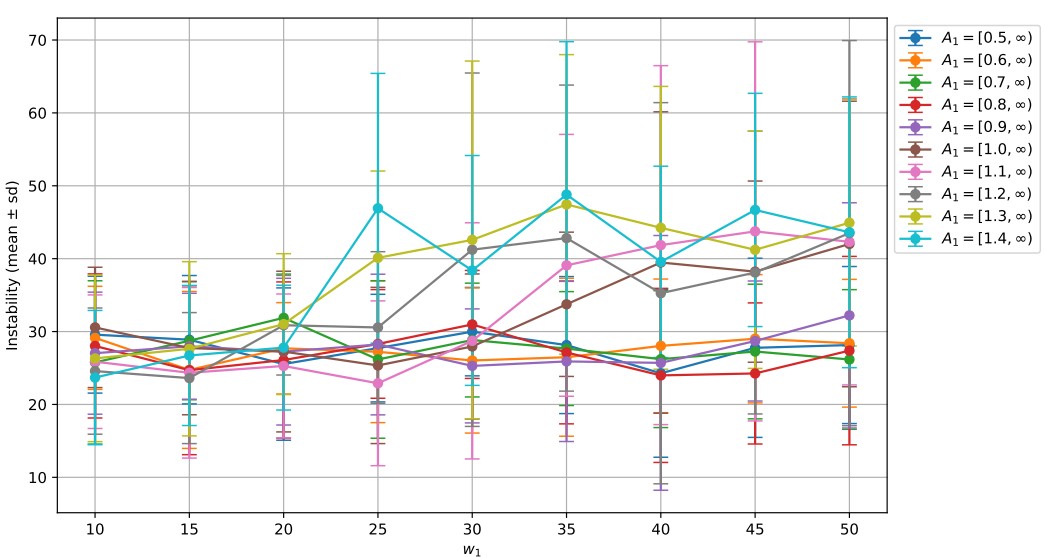

Figure 6: Influence of $w_1$ and $A_1$ on instability in sampling alpha-helix rich proteins.

evaluate the metrics helix% and instability. The results are displayed in Figures 5 and 6. We find that

- For a fixed $a_1$, as $w_1$ grows, the helix% of the generated sequences would grow at first, but may decline when $w_1$ is larger than some threshold. This threshold becomes smaller when $a_1$ gets larger.

- For small $a_1$, the instability index of the generated sequences does not vary significantly among different choices of $w_1$. But for large $a_1$, the generated sequences is prone to become more unstable when $w_1$ is large.

These results show that there may exist an optimal choice of the hyperparameters $w_1$ and $A_1$ that maximizes the helix% of the generated sequences while keeping low instability index, which may provide insights in designing the reward function $r(\cdot)$ for general controlled generation problems.

### B.3 PROTEIN SEQUENCES IN FIGURE 3.

From left to right, the uncontrolled generated sequences are:

1. RAGPRAPPRSDAGRTRGVGRKGQLLVTGKLDAPTLLSLPAAVKSTGATRS
2. MGFPNVPATAAPCPAAPTYEDYAAARGGSLPQVIQHALPVIFTAPLRKST
3. MNQQSTADIRMLIEIGSFMNDPNMMTLINLLILSNVFILLIVIYYRWRSL
4. MKFLARSTAKTEQLRERYLKTDIQILVYETIQGDFESIRLLPASVYNVSL
5. MLPSPAFAISEAQATVESGSIAGPELLAVAVEAPSTQDHRVFAGEETYGV

From left to right, the controlled generated sequences are

1. MINAEAADKDECRLADLLEAKELEMLELKALYLRLEEENKALKELARAMA
2. TACVEKPTHGNPTLHLAAKAFNAEIILDLAFLGQKREKLTQSNLRVISEK
3. MNNDEDLWWKSKGKLINKDKYKNLDNTIMYMKQNMKDIKELKGLETILNA
4. MNDQTREKLLKPGAAEVFAKKYRREKEAIESRAIARVADIDEALKLAAQL
5. MLAEPIGNIVTYAYVIILSILLLVKLGLAENMETSVALTTLLFSNIWQLR

These sequence are randomly sampled from the batch of generated sequences and are not cherry-picked.

### B.4 MORE DETAILS OF PROTEIN INPAINTING

The whole sequence of the protein is ERPLDVIHRSLDKDVLVILKKGFEFRGRLIGYDIHLNVVLADAEMIQDGEVVKRYGKIVIRGDNVLAISPT, where the cyan part with 35 amino acids is the prompt we use for inpainting. We choose the same hyperparameters as in Table 2. From left to right, the three sequences displayed in the blue box in Figure 4 are

1. MSLAVLKNSEDTLVKAELKGDVSVRGRLIGYDIHLNVVLADAEMIQDGEVVKRYGKIVIRGDSVVTVHLLTALESQIHEIEDEKAKADRAVKARTKAIKA
2. MEGIALKALMDFQVVMKLKGGKELRGRLIGYDIHLNVVLADAEMIQDGEVVKRYGKIVIRGTVITLIHIPEEVDFEAALKLLEKKPKKRIRRLKAEKSKK
3. SMSLAMQNLMGKEMKIRLAGGMCMRGRLIGYDIHLNVVLADAEMIQDGEVVKRYGKIVIRGNCIVYLDLPDSLKDELQSHERVHQYRGLKGAHAVKEKKR

## C CODES FOR ALGORITHMS 1 AND 2

```python
import tqdm
import torch
import numpy as np

@torch.no_grad()
def ctrl_gen(obtain_logits, obtain_reward, device,
            B, D, N, K, T,
            mask_state: int = None,
            invalid_ids: list = None,
            inpainting_pos: list = None,
            inpainting_values: list = None,
            gamma=lambda r: np.cos(r * np.pi / 2),
            prob_low_threshold=1e-10,
            disable_tqdm=False):
    """
    B: batch size
    D: sequence length
    N: vocabulary size including the masked token
    K: number of samples for Monte Carlo estimation
```

```
21
22      Sample from q(x) \\propto r(x) p(x), p(x) comes from the masked model,
            and r(x) is the reward function.
23
24      obtain_logits: [B,D] -> [B,D,N], return the logits P(X^d|X^UM)
25      obtain_reward: [*,D] -> [*], return the reward r(X)
26      return: [B,D], the sampled sequence
27
28      By default, mask_state is N-1, and the valid_ids (i.e., real tokens)
            are from 0 to N-2.
29      If there are other invalid tokens, one can specify the invalid_ids (
            default is [mask_state]).
30      The logits for invalid tokens will be set to -inf.
31      when inpainting_pos is not None, we will always inpaint the values at
            these positions with inpainting_values.
32      """
33
34      if mask_state is None:
35          mask_state = N-1
36      if invalid_ids is None:
37          invalid_ids = [mask_state]
38      elif mask_state not in invalid_ids:
39          invalid_ids = invalid_ids.append(mask_state)
40
41      if inpainting_pos is not None:
42          assert len(inpainting_values) == len(inpainting_pos)
43          assert all([0 <= pos < D for pos in inpainting_pos])
44          assert all([0 <= val < N and val !=
45                  mask_state for val in inpainting_values])
46          inpainting_values = torch.tensor(
47              inpainting_values, dtype=torch.int).to(device)
48
49      def inpaint(x):
50          """*,D -> *,D inpaint"""
51          if inpainting_pos is None:
52              return x
53          else:
54              shape = x.shape
55              x = x.reshape(-1, D)
56              x[:, inpainting_pos] = inpainting_values
57              return x.reshape(shape)
58
59      D_essential = D - len(set(inpainting_pos))
60      # only the dimensions in dims_to_sample will be sampled. The rest will
            be inpainted.
61
62      x = torch.full((B, D), mask_state, dtype=torch.int).to(device)
63      x = inpaint(x)
64      # B,D, initialize with mask_state and inpainted values
65
66      for t in tqdm(range(1, T+1), desc="Unmasking steps", disable=
            disable_tqdm):
67          if int(D_essential*gamma(t/T)) == int(D_essential*gamma((t-1)/T)):
68              continue # no more tokens to unmask in this step
69
70          # predict all the masked tokens
71          logits = obtain_logits(x) # B,D,N, p(x^d | x^UM)
72          logits[:, :, invalid_ids] = -np.inf
73          masked = x == mask_state # B,D
74          masked_logits = logits[masked] # ?,N
75          samples = torch.distributions.Categorical(logits=masked_logits).
                sample(
76              (K,)).to(dtype=torch.int, device=device) # K,?
77          x = x.repeat(K, 1).reshape(K, B, D).to(device) # K,B,D
78          x[:, masked] = samples
```

```
79          x = x.transpose(0, 1) # B,K,D
80          x = inpaint(x)
81          probs = obtain_reward(x) # B,K, p(y|x)
82          # avoid numerical instability during division
83          probs[probs < prob_low_threshold] = prob_low_threshold
84          weights = probs / probs.sum(dim=1, keepdim=True) # B,K, normalized
85          selected = torch.distributions.Categorical(probs=weights).sample()
                # B
86          x = x[torch.arange(B), selected] # B,D
87
88          # remask the tokens based on their confidence scores
89          confidence = torch.ones_like(x, dtype=torch.float64).to(device)
90          confidence[masked] = torch.rand_like(
91              confidence[masked], dtype=torch.float64).to(device)
92          low_k_values, low_k_indices = torch.topk(
93              confidence, k=int(D_essential*gamma(t/T)), dim=-1, largest=False
                  )
94          x[torch.arange(B).unsqueeze(1), low_k_indices] = mask_state
95      return x
```

