# OpenReview forum: "Plug-and-Play Controllable Generation for Discrete Masked Models"
_ICLR.cc/2025/Conference — ICLR 2025 Conference Withdrawn Submission_

### Official Review · Reviewer_kUek · 2024-10-20

**Soundness:** 2
**Presentation:** 3
**Contribution:** 1
**Rating:** 3
**Confidence:** 3

**Summary:**

This paper proposes a method to enable controllable generation given any unconditional generative model and a reward function as conditioning signal. This is done through computing importance weights using Monte-Carlo estimates and evaluates resulting samples using the given reward function. The authors demonstrate the effectiveness of their method using a toy dataset and in the context of conditional protein generation.

**Strengths:**

* The paper is generally well-written, and clear
* The method is relatively simple, and easy to implement
* The method only requires an unconditional model, and can be used to controllably generate from any conditional distribution given its corresponding reward function

**Weaknesses:**

* The novelty is relatively low, as importance sampling has been very well studied in prior works. Although to my knowledge, I have not seen it applied it in the context of controllable generation, the experiments do not well demonstrate the effectiveness of the proposed method
* Core experiments are on relatively easy (low-dim) distributions, and it is unclear as to how this method scales. How well does the method work for more complex distributions, e.g. for images, longer sequence proteins, etc? Do you need significantly more Monte Carlo samples?
* The method quite heavily relies on a good reward function -- which, in general may be difficult to properly specify. How does performance depend on how well the reward function is shaped?

**Questions:**

See Weaknesses

---

### Official Review · Reviewer_kZRj · 2024-11-01

**Soundness:** 3
**Presentation:** 4
**Contribution:** 2
**Rating:** 5
**Confidence:** 4

**Summary:**

The paper proposes a method for generating conditional samples from a masked generative model. Assuming the existence of a reward model, the method draw approximate samples from the unnormalized density r(x)p(x) without requiring the generative model to be retrained.

The method applies the Sampling Importance Resampling (SIR) trick to obtain approximate samples from the target distribution over the course of the generative process.

Experimental results demonstrate the concept on a toy problem as well as showcasing impressive results on a protein generation benchmark.

**Strengths:**

The paper presents a well-justified method from conditional sampling the presence of a reward function. It details the assumptions it makes and it gives an intuition when/why someone would use this method for conditioning.

In terms of novelty, SIR is not novel, but its application to masked generative models for controllable generations is. I am not aware of other works that use this idea for masked generative models.

The paper is very well written and easy to understand. The motivation is clear, the method is well-explained and detailed. Figure 1 and Algorithm 1 give an excellent overview that makes it easy to implement.

The experimental results on protein generation are extensive. They show convincing results on two benchmarks: solubility and alpha-helix percentage. Furthermore, they include a qualitative assessment of protein in-painting.

**Weaknesses:**

A main weakness of the paper is the experimental results. The work is motivated by the versatility of the approach: they claim strong performance across multiple domains. However, experimental results only include protein generation benchmarks. There are not experiments on text, images or audio with the masked models that are discussed in the introduction.

Regarding the protein benchmarks, there is no baseline to compare against and there are no ablation experiments.
* Baselines: It would be good to see how the method compares to naive fine-tuning approaches (while acknowledging that the proposed method is much lighter computationally).
* Ablations: The method does not have many hyperparameters to set, but it would be good to see how the generation quality depends on the number of Monte-Carlo samples used.

**Questions:**

Q: One of the motivations is that by setting r to be the p(y|x), one can sample from the Bayesian posterior p(x|y). How accurately can this method sample the Bayesian posterior and what kind of Bayesian inference problems can it be applied to?

The reasoning for my score is that I find the claims of effectiveness and versatility lack evidence.
* Effectiveness: The experiments have no baselines therefore its difficult to evaluate if the method is effective or not.
* Versatility: The method is only evaluated on a single domain (not counting the toy example).

I am willing to increase my score if the authors argue or provide further evidence in support of these two claims in the rebuttal.

---

### Official Review · Reviewer_MPEF · 2024-11-03

**Soundness:** 4
**Presentation:** 3
**Contribution:** 2
**Rating:** 6
**Confidence:** 3

**Summary:**

This model presents a method for controllable generation given a pre-trained (discrete) masking model without further training. Given a reward function, the method iteratively applies masking and remasking along with a mean-field approximation and importance sampling to perform controlled (e.g., conditioned on a class variable) generation in a "plug-and-play" manner. The work lays out the grounding theory and connections to (continuous and discrete) diffusion models, motivates the approach, and demonstrates the approach both on toy sequential data and protein generation (inpainting and class conditioned) tasks.

**Strengths:**

Overall the paper is very well written. The motivation of the problem, controllable discrete masked model generation without training, is good, as this implies flexible controllable generation without additional computational overhead of training for each controlled generation task. The theory appears to be sound to me without any errors, arriving at the mean field approximation with importance sampling, which seems to be a reasonable approach and yields decent results on both the toy task and protein generation tasks. The paper does an excellent job presenting the work as close to diffusion models, which makes the theory sections easy to read. The experiments for the most part are well motivated and the results do support the usefulness of the approach to some controllable settings.

**Weaknesses:**

No limitations are presented in the paper, and it seems like there may be some worth discussing. One is reward function design, as it's unclear whether some tasks may not have difficult to design reward functions or if there's a high dependence on reward function on success. The next is that the Monte Carlo samples seem to be quite high, the performance in figure 10 seems to indicate that even at 10k samples the model is still improving. There really should be more of a discussion about this limitation, which I believe is likely due to either the mean field approximation or the remasking schedule, but neither of these limitations / issues are discussed to any significant degree. Finally, I wonder why we're only looking at protein sequence generation as the task: why not also look at some natural language applications?

**Questions:**

1) Why did you choose to look only at protein sequences and not natural language for controllable generation tasks?
2) Is there a relationship between the number of MC samples needed and the mean field approximation or the remasking schedule? For example, if gamma is too high (or too low), does it take more samples to achieve high final reward?
3) What are some of the limitations of this model w.r.t. the reward design? What characterizes a controllable generative task for which reward design / success would be easy / hard?

---

### Official Review · Reviewer_rJsN · 2024-11-04

**Soundness:** 1
**Presentation:** 1
**Contribution:** 1
**Rating:** 1
**Confidence:** 4

**Summary:**

This paper tackles the problem of performing conditional generation of discrete structures via masked generative models. They propose a general-purpose approach for optimizing arbitrary objective functions during the generation process. Subsequently, they provide several simplifications and concrete modelling approaches to make the problem tractable and computationally efficient. Finally, they apply the methodology to a toy problem as well as a protein generation task.

**Strengths:**

* The paper tackles a broad category of problem; namely plug-and-play conditional generation using discrete masked models without the need for fine-tuning. Additionally, they lay out in which settings their methodology would be advantageous (for example, they indicate that this method is useful when evaluating the masked model is much more expensive to evaluate than the reward function).
* The authors make a good effort at making the paper reproducible by including source code of the algorithm (as well as detailed algorithm descriptions) in the appendix.
* Figure 1 is quite good and greatly facilitates understanding of the proposed methodology. In addition, the paper clearly describes the problem which they aim to solve, subsequently provides a concrete approach which makes the problem tractable, and performs some preliminary empirical validation.

**Weaknesses:**

__Theoretical Concerns__:

* Several key aspects of the paper lack a theoretical justification or are not derived in a principled manner. For example, the proposed reward equation $r(x) = \exp\left({-\sum w_i \text{dist}(m_i(x), A_i)^{\alpha_i}}\right)$ is provided with no theoretical grounding or explanation. As best I can tell, the definition of the sampling distribution $q(z) = Z^{-1} r(x)p(x)$ would require $r(x) \geq 0$ in order for $q(x)$ to be a valid distribution. However, this is not mentioned, and many alternative reward functions could be used. I would like to see a more detailed explanation of why this reward function was chosen (either theoretical justification or empirical results).
* Similarly, the use of the mean-field approximation and importance sampling present several practical challenges which are not addressed. In the case of importance sampling, the results are heavily dependent on samples obtained from regions of high density, and thus may require many monte-Carlo samples if the proposal distribution is far from the true distribution. Furthermore, the mean-field-approximation assumes that the probabilities of the masked inputs are independent conditioned on the observed values. This is clearly not the case for domains such as images, which exhibit strong local structure. The paper would be much improved with additional analysis of the performance of the proposed methodology when the assumptions are violated and/or on larger-scale problems more representative of real-world use.
* The authors mention that the proposed method is beneficial when the complexity of querying the masked model is much higher than evaluating the reward function. Unfortunately, this is only true for trivial objective functions. For example, protein structures are typically optimized for a complex objective that is computed by another deep learning model (i.e. predicting biological activity, folding structure, etc.). This calls in question the applicability of the method to wider categories of problems, as most problems of interest will not have a closed form/cheap objective function.

__Experimental Concerns__:
* In terms of the experimental validation, the experiments performed do not provide sufficient evidence that the methodology works as intended. First, the experiment using the toy problem uses a uniform masked model with a linear objective function. As expected, the proposed approach performs well given that the problem is explicitly formulated satisfy the mean-field approximation and importance sampling schemes. No attempt is made to characterize how the method performs as assumptions are violated. Furthermore, the protein experiments are conducted using objectives which are much too simple. GRAVY (Grand Average of hydropathy) is a simple sum of values per individual amino acid. Similarly, the instability index (Guruprasad et al., 1990), consists of summing values from a lookup table for pairs of amino acids. These objectives are simple enough that the assumptions of MFA and importance sampling are not violated, but are not representative in terms of computational costs or complexity of typical protein design tasks. Finally, an experiment is performed to optimize the helical fraction of the peptides. The objective used is not clearly defined in the paper, but validation is performed using ESM3. Consequently, if ESM3 is used for the helical fraction objective, then the objective would not be cheap to evaluate, and the initial assumptions made by the paper are violated. Overall, the paper would benefit from more extensive and principled empirical validation in settings more representative of how the methodology would be used in practice.
* Another aspect of the experimental results is that both the toy problem and the protein design task consist of relatively simple 1-dimensional discrete structures. I would need to see this methodology applied to more complex discrete structures such as 2D image generation or graph structures (such as per-atom molecule design) in order to validate some of the wider-scope claims made.
* In terms of presentation, many of the figures would benefit from more detailed captions to clearly present what is being shown. For example, figure 2 seems to imply that additional monte-carlo samples enable the algorithm to achieve a high degree of success when optimizing the objective, however this is only briefly touched upon in the main text, and not at all addressed in the caption. Additionally, figures 5/6 are quite visually crowded and hard to parse. As these figures occur in the appendix, the authors could take more space to make sure that the results are clearly and unambiguously presented.

__Contribution Concerns__
* The main contribution of the paper seems to be the introduction of the sampling distribution $Z^{-1} r(z)p(x)$, and then using MFA and importance sampling to sample from this distribution. This is not a novel methodology and is well known in various Bayesian settings. To accept the paper, there would need to be a more significant theoretical contribution. Additionally, there exists pre-existing plug-and-play samplers for continuous diffusion models, this paper extends plug-and-play samplers to discrete masked models, and this does not present a significantly novel framework for conditional generation.

**Questions:**

I have several questions regarding the content of the paper:
*  What was the metric used for computing/conditional generation when optimizing the helical fraction?
* How is the reward function on page 8 derived?  Additionally, why are the intervals for the metrics sometimes closed (i.e. instability with $A = [0, 40]$, and sometimes unbounded (i.e. helix % with $A = [0.8, \infty)$), and in what settings is bounded/unbounded preferable?
* Are the helical fractions correct in the protein experiment? In figure 3, the bottom two proteins seem almost identical, yet one has a helix fraction of 0.78, and the other 0.44. This does not seem quite correct.

---

### Note · Authors · 2024-11-28

I have read and agree with the venue's withdrawal policy on behalf of myself and my co-authors.